# 6-Oxofurostane and (iso)Spirostane Types of Saponins in *Smilax sieboldii*: UHPLC-QToF-MS/MS and GNPS-Molecular Networking Approach for the Rapid Dereplication and Biodistribution of Specialized Metabolites

**DOI:** 10.3390/ijms241411487

**Published:** 2023-07-14

**Authors:** Bharathi Avula, Ji-Yeong Bae, Jongmin Ahn, Kumar Katragunta, Yan-Hong Wang, Mei Wang, Yongsoo Kwon, Ikhlas A. Khan, Amar G. Chittiboyina

**Affiliations:** 1National Center for Natural Products Research, School of Pharmacy, University of Mississippi, University, MS 38677, USA; jybae@jejunu.ac.kr (J.-Y.B.); jmahn@kribb.re.kr (J.A.); kkatragu@olemiss.edu (K.K.); wangyh@olemiss.edu (Y.-H.W.); meiwang@olemiss.edu (M.W.); yskwon@kangwon.ac.kr (Y.K.); ikhan@olemiss.edu (I.A.K.); 2Division of Pharmacognosy, Department of BioMolecular Sciences, School of Pharmacy, University of Mississippi, University, MS 38677, USA

**Keywords:** phytochemicals, steroidal saponins, hyphenated methods, liquid chromatography, molecular networking

## Abstract

Identifying novel phytochemical secondary metabolites following classical pharmacognostic investigations is tedious and often involves repetitive chromatographic efforts. During the past decade, Ultra-High Performance Liquid Chromatography-Quadrupole Time of Flight-Tandem Mass Spectrometry (UHPLC-QToF-MS/MS), in combination with molecular networking, has been successfully demonstrated for the rapid dereplication of novel natural products in complex mixtures. As a logical application of such innovative tools in botanical research, more than 40 unique 3-oxy-, 3, 6-dioxy-, and 3, 6, 27-trioxy-steroidal saponins were identified in aerial parts and rhizomes of botanically verified *Smilax sieboldii*. Tandem mass diagnostic fragmentation patterns of aglycones, diosgenin, sarsasapogenin/tigogenin, or laxogenin were critical to establishing the unique nodes belonging to six groups of nineteen unknown steroidal saponins identified in *S. sieboldii*. Mass fragmentation analysis resulted in the identification of 6-hydroxy sapogenins, believed to be key precursors in the biogenesis of characteristic smilaxins and sieboldins, along with other saponins identified within *S. sieboldii*. These analytes’ relative biodistribution and characteristic molecular networking profiles were established by analyzing the leaf, stem, and root/rhizome of *S. sieboldii*. Deducing such profiles is anticipated to aid the overall product integrity of botanical dietary supplements while avoiding tedious pharmacognostic investigations and helping identify exogenous components within the finished products.

## 1. Introduction

The historical and traditional practices associated with herbal medicines have greatly influenced the isolation of various phytochemicals and their introduction into modern medicine to alleviate life-threatening diseases. Even though ethnopharmacological uses of the plants constituted one of the primary sources for the compounds focused on in the early stages of drug discovery and development, complex chemistries and tedious pharmacognostic investigations, together with reductionist, single-agent approaches, led to the limited success of phytomedicine in the modern world. Moreover, depending too heavily on supplementation with a single agent and ignoring the symbiotic natural chemical reservoir of phytochemicals in a given extract has adversely impacted botanical drug development. On the contrary, standardization and validation of plant sources used in clinical trials with herbal medicines, nutraceuticals, or dietary supplements have allowed others to replicate the results and been instrumental in establishing the scientific rigor for investigating the overall benefits of phytochemicals. 

Within the last two decades, untargeted metabolomic approaches such as UHPLC-HRMS/MS proved promising for generating a holistic view of sample chemical composition, aka chemical fingerprints, without focusing on any specific chemical class. The resulting high-performance chromatographic data or chemical profiles were successfully demonstrated for their utility in establishing botanical species identification, authentication, and confirmation of the quality of the botanical extracts in various finished products without the need for time-consuming isolation of every phytochemical present in a given matrix. Moreover, chromatographic methods coupled with mass spectroscopy and the implementation of tools such as global natural products social molecular networking (GNPS) revolutionized the rapid dereplication of natural products. 

In the quest to ensure the quality and safety of botanical ingredients in various matrices, we recently demonstrated that 5-hydroxylaxogenin in many dietary supplements is from poor-quality synthetic sources [1] rather than being a phytochemical constituent of *Smilax sieboldii* as touted. *Smilax sieboldii* Miq (Smilacaceae), a deciduous climbing shrub, is a key member of 350 species widely distributed in Republic of Korea, China, and Japan. The young, tender leaves of this plant are used as an edible vegetable, and the underground root/rhizomes have been used to treat various ailments in Republic of Korea [2] and other traditional practices [3,4,5]. Even though many steroidal saponins of different structural types, viz., (iso)spirostane, furostane, cholestane, and pregnane type, were isolated from the *Smilax* genus [6], only 11 compounds representing sapogenins and saponins from the stem and underground parts of *S. sieboldii* were reported [7,8,9]. In general, the sugar moieties in these saponins are linear or branched saccharide chains made up most often of glucopyranosyl (Glc) and arabinopyranosyl (Ara) moieties at C-3 and C-26 positions appended via an ether linkage. Okanishi et al. [9] obtained laxogenin as one of the main sapogenins in the stem of *S. sieboldii*, and Kim et al. [10] reported laxogenin contents as 0.059% (*w*/*w* from dried material) in this plant using HPLC with refractive index (RI) detection. The extracts and secondary metabolites from *S. sieboldii* were reported to possess anti-adipogenic activity on 3T3-L1 adipocytes [11] and anti-inflammatory activity on LPS-activated NO production in macrophages [12]. Nevertheless, in order to gauge the applicability of the untargeted metabolomics approach in quality assurance of botanicals, the leaf, stem, and root/rhizome of botanically verified *S. sieboldii* were investigated using UHPLC-QToF-MS/MS and GNPS platforms. Specifically, the characteristic mass spectral properties of two distinct aglycone reference standards, diosgenin and laxogenin, were utilized to develop the plant-part-specific, holistic steroidal composition in *S. sieboldii*.

## 2. Results and Discussion

Full-scan MS and MS/MS spectral data were utilized to establish each analyte’s accurate mass, including their tentative chemical formula, using QToF. The Agilent Molecular Features Extractor (MFE), which resolves the co-eluting interferences and groups the isotopic clusters (adducts, dimers, and trimers, if any), was applied to facilitate the conversion of the large volume of mass spectral data into a set of possible compounds. 

Liquid chromatography-electrospray ionization-mass spectrometry (LC-ESI-MS) was optimized to resolve steroidal saponins in methanolic extracts of *S. sieboldii*. Both positive and negative ESI ionization modes were investigated with UHPLC-QToF and used to establish the total ion chromatogram (TIC) of saponins for each extract. The proposed molecular formulas and the identity of possible compounds were established based on accurately calculated masses and fragment ions from the corresponding aglycones. The dominant fragmentation pathways of steroidal saponins were investigated in positive ESI mode. The MS and MS/MS spectral data for all forty-one (**1**–**41**) analytes detected in negative and positive ESI modes are outlined in Table 1.

### 2.1. Optimization of Conditions for UHPLC-QToF-MS

A Poroshell 120-EC8 column was identified to have a superior resolution for most peaks considering the complexity of the sample associated with various saponins. The column packing has a solid core of 1.7 μm with a 0.5 μm thick porous outer layer and a total particle size of 2.7 μm, providing high efficiency at lower column pressures with a flow rate of 0.25 mL/min. Adding 0.1% of formic acid to the mobile phase alleviates peak tailing and improves overall ionization. The compounds are well detected in both positive and negative ion modes. In addition to establishing the characteristic retention time (Rt) indices, a standard 5 ppm threshold yielded an accurate mass measurement of elemental composition for each analyte’s precursor and fragment ions. The fragmentor voltage, the voltage applied to the exit of the capillary to introduce ions into the mass spectrometer (a high-pressure region), usually cannot be fixed for each compound independently due to the proximity of other targets. The collision energy, 50 eV, furnished key fragment information for the structural assignment of both furostane and (iso)spirostane types of saponins. In the case of isobaric analytes with identical MS/MS spectra, characteristic retention indices were found reliable for identification purposes. Based on these critical parameters and spectral properties, (iso)spirostane-type saponins were identified as major metabolites. The specific fragmentation patterns of reference standards (diosgenin and laxogenin) were investigated and were instrumental in identifying steroidal saponins in these extracts according to retention times, MS, and MS/MS data. Both [M-H]^−^ and strong [M+HCOO]^−^ (due to formic acid in the mobile phase) ions were noticed under negative-ion ESI-MS mode (Table 1). In comparison, [M+H]^+^ and [M+Na]^+^ ions (Table 1) with extractive ion chromatography (EIC) together with the loss of each sugar unit as a result of glycoside bond cleavage were observed in positive-ion acquisition mode. Mass spectral data associated with the sequential loss of sugar units proved very valuable in assembling information on the type of aglycone and the number and types of sugar units present in a given analyte’s structural arrangement. The sugar analysis was carried out based on the published method [13] and found to contain glucose and arabinose. The structures of the aglycones of the isolated saponins of *S. sieboldii* primarily belong to the spirostane/isospirostane type, a hexacyclic ABCDEF-ring system. Most of these compounds are furostane-type (**8**–**16**, **18**–**22**, **27**–**35**, and **37**–**41**), a pentacyclic ABCDE-ring system with the sixth open F ring, and precursors of spiro/isospirostane-type saponins (**1**–**7**, **17**, **23**–**26**, and **36**) (Figure 1). The (iso)spirostane saponins’ structures are derived from diosgenin, laxogenin, or sarsasapogenin/tigogenin (*m*/*z* 415, 431, and 417) as backbone aglycones. 

### 2.2. Unified Fragmentation Pattern with Isospirostane-Type Saponins

Isospirostane-type saponins are monodesmosidic glycosides with six rings (A–F) in the sapogenin. These types of compounds are typically characterized by the presence of equatorial (hydroxy)methyl at the C-25 position on the F-ring. In contrast, the methyl group at C-25 is usually in the axial position for spirostane-type glycosides. Nevertheless, these isospirostanes are further classified into subtypes based on aglycone backbone structures: diosgenin (**1a**), laxogenin (**1b**), and sarsasapogenin/tigogenin (**1c**), which are functionalized at C-5, C-6, and C-22 or C-27 positions. For example, a double bond between C-5 and C-6 in the diosgenin-type, an oxo-group at C-6 for a laxogenin-type, and a saturated single bond between C-5 and C-6 positions in the sarsasapogenin/tigogenin-type aglycones are usual, along with their existence as furostane-type (hemiketal) or (iso)spirostane-type (ketal) structural arrangements where the sugar moieties are appended at the C-3 and/or C-26 positions. 

Before delving into clusters and fragmentation of complex saponins, mass spectral features of three aglycones, diosgenin, laxogenin, and sarsasapogenin/tigogenin, were investigated. Highly cohesive fragmentation patterns were observed for all three aglycones, in which either loss of water or C_8_H_16_O_2_ were observed as initial fragments, and subsequent loss of the C_8_H_16_O_2_ moiety or water resulted in a common fragment ion. Further dehydration resulted in an ion with [M+H-C_8_H_20_O_4_]^+^ independent of functionalization at the C-5 and C-6 positions, as shown in Figure 1. Based on these mass spectral features, at least five unique clusters were identified (as outlined below), in which several isobaric and anomeric isomers were tentatively identified and compared with reported data, if applicable [8].

#### 2.2.1. Cluster #1: Laxogenin-Type Saponins (**1**–**16**)

Based on mass spectral data, including MS/MS data (Table 1, Appendix A) and molecular networking analysis, a cluster of saponins (**1**–**16**) representing a laxogenin-type aglycone (*m*/*z* 431) as a common backbone was observed. Additionally, all these saponins resulted in four key fragments (*m*/*z* 413.3037 [M+H-H_2_O]^+^, 287.1998 [M+H-C_8_H_16_O_2_]^+^, 269.1894 [M+H-C_8_H_16_O_2_-H_2_O]^+^, and 251.1789 [M+H-C_8_H_16_O_2_-2H_2_O]^+^) by consecutive loss of (iso)spirostane, methylated pyran, and water units, respectively, which are characteristic fragments of a protonated laxogenin molecule with *m*/*z* 431.3161. Compounds **2**–**3** were isomers of each other with MWs of 724 Da (di-glycosides) and identical fragmentation pathways with two different retention indices. Both compounds produced a deprotonated molecule [M-H]^−^ (*m*/*z* 723.3958) and a strong [M+HCOO]^−^ ion at *m*/*z* 769.4012 in negative ion mode and *m*/*z* 725.4099 in positive ion mode. The protonated molecule [M+H]^+^ (*m*/*z* 725) in (+)-ESI-MS gave seven major fragments (*m*/*z*) by the consecutive loss of one arabinosyl, one glucosyl, and water moieties from the molecular ion *m*/*z* 725.4099. In (+)-ESI-MS/MS, the mass spectrum was dominated by the characteristic fragment ion [M+H-Ara-Glc]^+^ (*m*/*z* 431.3159). Indeed, isobaric smilaxin A and laxogenin 3-*O*-α-L-arabinopyranosyl-(l→6)-β-D-glucopyranoside are reported to be constituents of *S. sieboldii* [8] and *S. lebrunii* [14]. 

Compounds **4**–**7** and **8**–**9** were tentatively identified as tri-glycosidic isomers with MWs of 886 and 904 Da, respectively. Saponins representing **4**–**7** possessed an identical [M+H]^+^ with *m*/*z* 887.4646, resulting in seven major fragments with *m*/*z* 725.4094, 593.3676, 431.3150, 413.3035, 287.1988, 269.1883, and 251.1764 as a result of the consecutive loss of two glucose, one arabinose, 144 Da (formula C_8_H_16_O_2_), and water molecules from the parent ion. Furthermore, two ions corresponding to [M-H]^−^ at *m*/*z* 885.4483 and [M+HCOO]^−^ at *m*/*z* 931.4554 in negative ion mode enabled the identification of (iso)spirostane-type tri-glycosides with a laxogenin aglycone motif. The existence of similar compounds, smilaxin B or isobaric laxogenin glycoside, has already been reported in *Smilax* [8] and *Allium* spp. [15]. Compounds **8**–**9** resulted in an *m*/*z* 887 molecular ion as [M+H-H_2_O]^+^ and eight main fragments (*m*/*z* 755.4205, 725.4087, 593.3677, 431.3157, 413.3051, 287.2002, 269.1894, and 251.1790) attributed to the consecutive loss of two glucosyls, one arabinosyl, 144 Da (formula C_8_H_16_O_2_), and water molecules from the molecular ion at *m*/*z* 887.4628. The ions corresponding to *m*/*z* 903.4579 [M-H]^−^ and 949.4645 [M+HCOO]^−^ and the fragment ions at *m*/*z* 771.4161 [M-H-Ara]^−^, 741.4065 [M-H-Glc]^−^, and 609.3614 [M-H-Ara-Glc]^−^ confirmed furostane-type tri-glycosides with a laxogenin backbone [8]. 

Compounds **10**–**12** produced an identical deprotonated molecule [M-H]^−^ at *m*/*z* 1065.5112 and a strong [M+HCOO]^−^ ion at *m*/*z* 1111.5167. In positive ion mode, the ion with *m*/*z* 1049 [M+H-H_2_O]^+^ was observed with nine major fragments by the consecutive loss of one arabinose, three glucose units, and water molecules from the molecular ion *m*/*z* 1049.5184. In (+)-ESI-MS/MS, the mass spectrum was dominated by the characteristic fragment ion [M+H-H_2_O-Ara-3Glc]^+^ (*m*/*z* 431.3162). The fragment ion at *m*/*z* 887.4619 could be attributed to the loss of 162 Da (formula C_6_H_10_O_5_) as the terminal unit [M+H-H_2_O-Glc]^+^ from the dehydrated protonated molecule. Based on these mass spectral features, saponins representing **10**–**12** were tentatively assigned as tetra-glycosides of laxogenin. Such types of spirostanol saponins are reported to be constituents within the bulbs of Chinese onion (*Allium chinense*) [16] and *Lilium callosum* [17]. 

Similarly, compound **13**, having *m*/*z* 1181.5577 [M+H-H_2_O]^+^, and saponins representing **14**–**16** with *m*/*z* 1228 Da, were tentatively assigned as furostanol-type, bisdesmosidic penta-glycosides wherein an extra hexose (arabinose for **13** and glucose for **14**–**16**) was attached at the C-26 position due to labile loss of water and hexose units. After the initial loss of water and hexose, these analytes also furnished identical fragment ions corresponding to the loss of sugar moieties, C_8_H_16_O_2_, and water molecules. Furostanol-type saponins, similar to asparasaponins I and II from the shoots of *Asparagus officinalis* [18] and chinenoside I from bulbs of *Allium chinense* (Liliaceae) [19], were isolated and fully characterized. The comprehensive fragmentation pathways associated with these laxogenin-type saponins are delineated in Appendix A. 

#### 2.2.2. Cluster #2: Compounds **17**–**22**

According to the MS and tandem mass data (Table 1, Appendix A), in comparison with the diosgenin(**1a**) fragmentation pattern, saponins **17**–**22** were tentatively deduced to have the same aglyconic backbone as the diosgenin with *m*/*z* 415. Compound **17** was deduced as a di-glycoside of diosgenin based on protonated molecular ions at *m*/*z* 709.4170, fragment ions at *m*/*z* 577.3738 (loss of arabinose), and *m*/*z* 415.3204 (loss of arabinose and glucose), and subsequent fragmentation of *m*/*z* 415 to furnish the same daughter ions as diosgenin at *m*/*z* 397, 271, and 253. Except for the hydroxyl group at C-27, similar glycosides were also reported to be constituents of *S. lebrunii* [20] and *S. scobinicaulis* [21]. 

Compounds **18**–**19** were identified to possess the same molecular ions with *m*/*z* 871.4686 due to the loss of the water molecule [M+H-H_2_O]^+^. In addition to water loss, fragments (*m*/*z* 739.4221 [M+H-H_2_O-Ara]^+^, 709.4157 [M+H-H_2_O-Glc]^+^, 577.3738 [M+H-H_2_O-Ara-Glc]^+^, 415.3204 [M+H-H_2_O-Ara-2Glc]^+^, 397.3097 [M+H-2H_2_O-2Glc-Ara]^+^, 379.2989 [M+H-3H_2_O-2Glc-Ara]^+^, 271.2044 [M+H-H_2_O-2Glc-Ara-C_8_H_16_O_2_]^+^, and 253.1950 [M+H-2H_2_O-2Glc-Ara-C_8_H_16_O_2_]^+^) corresponding to the consecutive loss of two glucose units, one arabinose unit, a 144 Da (C_8_H_16_O_2_) fragment, and water molecules were also observed in **18**, **19**. Hence, these compounds were tentatively established as diastereomeric furostane-type bisdesmosidic tri-glycosides. A similar furostane-type glycoside with an additional hydroxy group at the C2 position has been reported to be a constituent of fenugreek by UHPLC-MS [22]. Compounds **20**–**22** were identified with an isobaric molecular ion [M+H-H_2_O]^+^ at *m*/*z* 1033.5226 and fragment ions (loss of sugars, 144.115 Da, and water molecules) identical to di- and tri-glycosides of diosgenin (**1a**) except with an additional glucose moiety attached at the C-3 position. At least eight other sapogenins with different sugar units (rhamnose instead of arabinose) appended at the C-3 position were isolated and characterized from *Allium cepa* L. var. *tropeana* seeds, highlighting the natural existence of such furostane-type glycosides [23]. These compounds were tentatively identified as furostane-type tetra-glycosides with a glycoside group with two glucose units, one arabinose unit at C-3, and one glucose unit at the C-26 position. The comprehensive fragmentation pathways associated with these diosgenin-type saponins are delineated in Appendix A.

#### 2.2.3. Cluster #3: Compounds **23**–**33**

Like the other two molecular networks of laxogenin and diosgenin saponins, another cluster of components shared the same aglycone, sarsasapogenin/tigogenin, as a backbone structure. Based on mass spectral features under positive and negative ionization modes together with loss of fragments (sugar units, 144.115 Da, water molecules), compound **23** with *m*/*z* 711.4299 [M+H]^+^ was tentatively identified as a di-glycoside of sarsasapogenin/tigogenin; 3-*O*-arabinoglucoside, and compounds **24**–**26** with *m*/*z* 873.4834 [M+H]^+^ were tentatively identified as tri-glycosides of sarsasapogenin/tigogenin; 3-*O*-arabinodiglucosides. Many structurally similar compounds functionalized at the C-2 position or appended with different sugar moieties at C-3 have been reported to be constituents of *Trigonella foenum-graecum* seeds [24]. Other sets of tri-glycosides attached to a furostane-type motif with *m*/*z* 873.4834 [M+H-H_2_O]^+^ or *m*/*z* 889.4802 [M-H]^−^ (for **27** and **28**) and *m*/*z* 903.4943 [M+H-H_2_O]^+^ or *m*/*z* 919.4906 [M-H]^−^ (for **29** and **30**) were also present in *S. sieboldii* extract, the only difference being the type of sugar units appended at the C-3 position, i.e., the former saponin with arabino-glucoside and the latter with di-glucoside. In addition to these saponins, tetra-glycosides, **31**–**33** with *m*/*z* 1035.5391 [M+H-H_2_O]^+^, were also tentatively identified as other saponins with sarsasapogenin/tigogenin as a backbone. These are mono-desmosidic steroidal saponins containing hexose moieties at the C-3 position. The comprehensive fragmentation pathways associated with these tigogenin-type saponins are delineated in Appendix A.

#### 2.2.4. Cluster #4: Compounds **34**–**35**

According to the MS and tandem mass data (Table 1, Appendix A), compounds **34**–**35** with *m*/*z* 889.4792 [M+H]^+^ (C_44_H_72_O_18_) were tentatively deduced as (iso)spirostane-type tri-glycosides derived from a common backbone, a 3, 27-dihydroxy furostane structure, as shown in Appendix A. Many of these saponins with functionalization at C-3 and C-27 have already been reported to be constituents identified in *Smilax* spp. [25]. The cohesive fragmentation pattern further confirmed that these two compounds are diastereomers, with a trisaccharide unit having two glucose and one arabinose appended at the C-3 position of the triterpene. Like other saponins, consecutive losses of sugars, 142.0994 Da (C_8_H_14_O_2_), and water molecules were observed from the parent ion. The exact cohesive fragmentation pattern is outlined in Appendix A. 

#### 2.2.5. Cluster #5: Compounds **36**–**40**

Using mass fragmentation data together with tandem mass features (Table 1, Appendix A), compounds **36**–**40** were deduced from the same backbone aglycone with *m*/*z* 447 in positive ion mode and shared identical fragmentation pathways. Considering the elimination of 160 Da, which is 16 Da larger than the characteristic elimination of the C_8_H_16_O_2_ unit with 144 Da (observed for all other saponins mentioned above), it is reasonable to establish that these compounds **36**–**40** have a hydroxyl substituent at the C-27 position and are identical to laxogenin-type compounds **2**–**16** mentioned under Section 2.2.1. On the basis of (+)-ESI-MS data and main fragments, compound **36** was tentatively identified as 3, 27-dihydroxy-6-keto (iso)spirostane, **37** and **38** as (iso)spirostane-based di-glycosides with the molecular formula C_38_H_60_O_14_, *m*/*z* 741.4063 [M+H]^+^, and compounds **39** and **40** as tri-glycosidic (iso)spirostane-type saponins with C_44_H_70_O_19_, *m*/*z* 903.4536 [M+H]^+^. 

In addition to 27-hydroxy saponins **36**–**40** (Appendix A), compound **41** was tentatively identified with a molecular ion [M+H-H_2_O]^+^ (*m*/*z* 1067.5268). Based on the tandem mass fragmentation pattern and observed fragment ions (*m*/*z* 935, 905, 773, 743, 725, 611, 593, 449, 431, 413, 395, 287, 269, and 251) due to sequential losses of either sugar units or hydroxyl groups and 144.115 Da, this compound was tentatively assigned as a tetra-glycoside of 3, 6, 27-trihydroxy furostane-type saponin (Appendix A). Such saponins have already been reported as constituents of *S. scobinicaulis* [26] and *S. sieboldii* [7,8]. 

### 2.3. Biodistribution of Saponins within S. sieboldii Plant Parts

Steroidal saponins lack a remarkable chromophore for UV and visible detection; therefore, MS was employed for detection in UHPLC analysis. In addition, MS offers a much greater selectivity to discern possible interferences. Optimized chromatographic conditions were achieved after several trials with acetonitrile, formic acid, and water in different proportions as the mobile phase. To assess the biodistribution of various saponins, nine samples of the root, leaf, or stem originating from *S. sieboldii* were used. To rapidly dereplicate the distribution of saponins in various parts of *S. sieboldii*, the UHPLC-MS/MS data of the extracts were analyzed using the GNPS-based molecular network method as enabled through the Cytoscape platform “https://cytoscape.org/ (accessed on 9 July 2022)”. Laxogenin (**1b**), previously reported as a constituent of *S. sieboldii* root, and diosgenin (**1a**) were utilized as seed molecules to probe the corresponding glycosides and related sapogenins present in *S. sieboldii*. 

The molecular networking analysis revealed at least 92 spectral nodes with mass spectral features of the seed molecule laxogenin. Subsequent analysis of 92 nodes resulted in the identification of three clusters with multiple spectral nodes representing di-, tri-, tetra-, and penta-saccharides of 6-keto (iso)spirostane or its precursor, 6-keto furostane, derived from one to two arabinose and two to four glucose moieties (Figure 2 and Appendix A). To substantiate the network analysis further, the corresponding four 6-keto furostanes, oxygenated at the C-27 position (**36**–**40**, Appendix A), and a 3, 6, 27-trihydroxy furostane-type sapogenin (Appendix A) were also detected in *S. sieboldii* based on mass spectral features of the seed molecule laxogenin. In addition, to these 6-position functionalized sapogenins, another cluster representing diosgenin as the backbone with *m*/*z* 415 was identified through molecular network analysis. Further analysis of nodes corresponding to diosgenin resulted in the identification of di- and tri-glycosides consisting of one arabinose and 1–2 glucose units attached at the C-3 position and furostane-type tetra-glycosides with a glycoside group with one arabinose and two glucose units at C-3 and one glucose unit at the C-26 position (Appendix A). Moreover, one large cluster representing either sarsasapogenin or tigogenin-like sapogenins was identified with molecular network analysis. Even though several sarsasapogenin-like compounds were isolated from the *Smilax* genus, the mass spectral data could not distinguish between a *cis*- or *trans*-fusion between rings A and B, necessitating further pharmacognosy investigations to establish these compounds as sarsasapogenin-like or tigogenin-like sapogenins (Appendix A). Nevertheless, at least 11 nodes representing sarsasapogenin- or tigogenin-like sapogenins were identified within the cluster. Moreover, a cluster representing two oxygenated compounds (**34** and **35**, Appendix A) belonging to 27-hydroxy sarsasapogenin/tigogenin-like furostane glycosides was also identified through network analysis. In summary, by analyzing these clusters based on their HRMS spectra and the corresponding MS/MS fragmentation pattern, 41 nodes were deduced to be different sapogenins in various parts (leaf, stem, or root) of *S. sieboldii*. The biodistribution of these analytes (**1**–**41**) according to their plant parts is tabulated in Table 1. 

To assure the existence of these saponins and avoid false positives, mass spectral data of all 41 compounds were manually inspected and tentatively categorized primarily as di-, tri-, tetra-, and penta-saccharides derived from either diosgenin, laxogenin, or sarsasapogenin/tigogenin backbone structures. Importantly, through the implementation of GNPS and MS/MS spectral data, two other aglycones, 3, 27-dihydroxy furostane and 3, 6, 27-trihydroxy furostane, were identified as novel backbones in some of the saponins identified in *S. sieboldii*. Indeed, these closely related furostanes (Figure 3) might be two important missing building blocks in the biogenesis of laxogenin-type saponins in many *Smilax* spp. Understanding the roles of enzymes associated with the biogenesis of oxidative furostane-type saponins and establishing their biological activities makes *S. sieboldii* an interesting botanical for further study.

## 3. Materials and Methods

### 3.1. Standards and Chemicals

Laxogenin from Proactive Molecular Research (Alachua, FL, USA) and diosgenin from MP Biomedicals Inc. (Irvine, CA, USA) were purchased, and their purity was confirmed by spectral data (1D- and 2D-NMR, ESI-HRMS). HPLC-grade solvents (acetonitrile and methanol) and formic acid were purchased from Fisher Scientific (Fair Lawn, NJ, USA). Water for the mobile phase was purified using a Milli-Q system (Millipore, Burlington, MA, USA).

### 3.2. Plant Materials

Ten samples, including dried leaf (#20801, 20804, 20807), stem (#20802, 20805, 20808), root (#20803, 20806, 20809), and whole plant (stem, leaf, root) (#20836) of *S. sieboldii*, were sourced from three different locations in Republic of Korea (Chuncheon, Yanggu, and Yeongwol). The botanical authenticity of these samples was verified by Prof. Yongsoo Kwon (College of Pharmacy, Kangwon National University, Chuncheon, Republic of Korea). Specimens of all samples are deposited at the NCNPR botanical repository, the University of Mississippi, University, MS, USA.

### 3.3. Plant Sample and Standard Preparations

Standard compounds diosgenin and laxogenin were prepared at a 1 mg/mL concentration and further diluted to achieve a 10 μg/mL concentration of each analyte. 

Each dried, powdered plant sample (1 g) was suspended in methanol (5 mL) and sonicated for 30 min, followed by centrifugation at 959× *g* for 15 min. The resulting supernatant was transferred to a 10 mL volumetric flask. The procedure was repeated twice, and the respective supernatants were combined. The final volume was adjusted with methanol to 10 mL and mixed thoroughly. Before injection, an adequate volume (ca. 2 mL) was passed through a 0.45 μm PTFE membrane filter. The first 1.0 mL was discarded, and the remaining volume was collected in an LC sample vial for further analysis.

### 3.4. Ultra-High Performance Liquid Chromatography Coupled to Quadrupole Time of Flight-Mass Spectrometry (UHPLC-QToF-MS/MS)

The liquid chromatographic system was an Agilent Series 1290 comprised of the following modular components: a binary pump, a vacuum solvent micro degasser, an autosampler with a 100-well tray, and a temperature-controlled column compartment. An Agilent SB-C8 (100 × 2.1 mm ID, 1.8 μm) column with the mobile phase consisting of water (A) and acetonitrile (B) containing 0.1% formic acid at a flow rate of 0.23 mL/min was utilized in the current method. The column temperature was set at 40 °C. The following gradient: 0 min, 97% A: 3% B to 30% B in 15 min, next 10 min to 50% B, next 10 min to 70% B, and then to 100% B in the next 5 min was implemented to achieve optimal separation. Each separation was followed by a 5 min washing with 100% B and a 5 min re-equilibration period before injecting a new sample. Three microliters of the sample were injected. 

The mass spectrometric analysis was performed with a QToF-MS/MS (Model #G6530A, Agilent Technologies, Santa Clara, CA, USA) equipped with an ESI source with Jet Stream technology connected with a nitrogen generator (Peak Scientific, Inchinnan, UK) using the following parameters: drying gas (N_2_) flow rate, 9 L/min; drying gas temperature, 300 °C; nebulizer, 30 psig; sheath gas temperature, 300 °C; sheath gas flow, 9 L/min; capillary, 3500 V; skimmer, 65 V; Oct RF V, 750 V; and fragmentor voltage, 125 V. The sample collision energy was set at 50 eV. The total analysis time was 40 min. ESI-QToF analysis was performed in a 2 GHz extended dynamic range positive ionization mode. The identification of compounds present in samples was compared with reference standards (diosgenin and laxogenin). The tentative identification of some derivatives was based on an accurate or exact mass spectrum.

All the operations, acquisition, and data analysis were controlled by Agilent MassHunter Acquisition Software v. A.05.00. Each sample was analyzed in positive ion mode in the *m*/*z* 50–1500 range. Accurate mass measurements were obtained using ion correction techniques with reference masses at *m*/*z* 121.0509 (protonated purine) and 922.0098 [protonated hexakis (1H, 1H, 3H-tetrafluoropropoxy) phosphazine, or HP-921] in positive ion mode. The compounds were confirmed in each spectrum. For this purpose, the reference solution was introduced into the ESI source via a T-junction using an Agilent Series 1200 isocratic pump (Agilent Technologies, Santa Clara, CA, USA) with a 100:1 splitter set at a flow rate of 20 μL/min. The LOD (limit of detection) was between 100 and 500 ng/mL for all reference compounds analyzed.

MassHunter Workstation software (version 10.0), including Qualitative Analysis (version B.07.00), was used to process raw MS and MS/MS data, including background subtraction, filtering, molecular feature extraction, and molecular formula estimation. To subtract molecular features (MFs) originating from the background, a blank sample (methanol) was analyzed under identical instrument settings, and background MFs were removed. MFs were characterized by retention time, the intensity at the apex of the chromatographic peak, and an accurate mass. The voluminous data were converted using Agilent Molecular Features Extractor (MFE), which resolved co-eluting interferences and grouped the isotopic clusters, all adducts, dimers, and trimers, if present.

### 3.5. Global Natural Products Social Molecular Networking (GNPS) Analysis

The MS/MS data of the extracts were converted to .mzXML format files using MS-Convert software from ProteoWizard [27] and then uploaded on the GNPS (global natural products social molecular networking) web platform “http://gnps.ucsd.edu (version 1.3.16-GNPS)” to determine molecular networks. Parameters for the molecular network generation were set as follows: precursor mass tolerance *m*/*z* 0.02 Da, MS/MS fragment ion tolerance *m*/*z* 0.02 Da, minimum cosine score 0.6, minimum matched fragment ions 3, and minimum cluster size 1. Further edges between two nodes were kept in the network if each node appeared in the other’s top 10 most similar nodes. The spectral library matching was performed similarly to the input data [28], and the resulting molecular networks were visualized using Cytoscape 3.7.0 software. The putative identification of steroid glycosides was performed by manual interpretation of MS/MS spectral data.

## 4. Conclusions

In summary, a UHPLC-QToF-MS technique has been applied to characterize steroidal saponins from *S. sieboldii*, avoiding the time-consuming derivatization steps and tedious purification of compounds from the crude extracts. The characteristic fragmentation patterns observed in QToF-MS/MS spectra allow the identification of aglycones and the number of sugar moieties. A total of forty-one steroidal saponins with six aglycone skeletons were characterized in methanolic extracts of *S. sieboldii* to possess various sugar units (glucose and arabinose) that were appended to the triterpene’s C-3 position via an ether linkage. These contain mostly furostane-type and (iso)spirostane-type compounds. Laxogenin and diosgenin were used as reference standards. In this work, an UHPLC-QToF-MS analytical method has been successfully demonstrated to characterize steroidal saponins. MS in negative ion mode for most compounds resulted in prominent [M-H]^−^ or [M+HCOO]^−^ molecular ions. At the same time, acquisition in positive ion mode furnished key structural information for the tentative elucidation of these phytochemicals. 

Moreover, implementing an LC-MS/MS-based molecular networking approach has been proven to be a powerful method for rapidly dereplicating phytochemicals, specifically minor constituents in complex mixtures. A traditional UHPLC/ESI-QToF-MS/MS analytical tool coupled with GNPS allowed for the identification of at least six aglycone skeletons (Figure 3) as the backbone structures for these saccharides, with the majority distributed throughout the entire plant, independent of the plant part analyzed. Nevertheless, the analytical approach presented here would be an ideal tool to establish a “biogenetic molecular network fingerprint” and to aid in plant-part-specific or species-specific network fingerprints. Additionally, by comparing such network profiles in commercial supplement products, one could determine the quality of *Smilax*-based natural extracts used in various finished products and identify any intentional or unintentional adulteration with synthetic steroids.

## Figures and Tables

**Figure 1 ijms-24-11487-f001:**
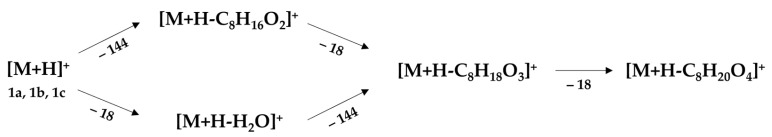
General cohesive fragmentation pattern associated with diosgenin (**1a**), laxogenin (**1b**), and sarsasapogenin/tigogenin (**1c**) aglycones.

**Figure 2 ijms-24-11487-f002:**
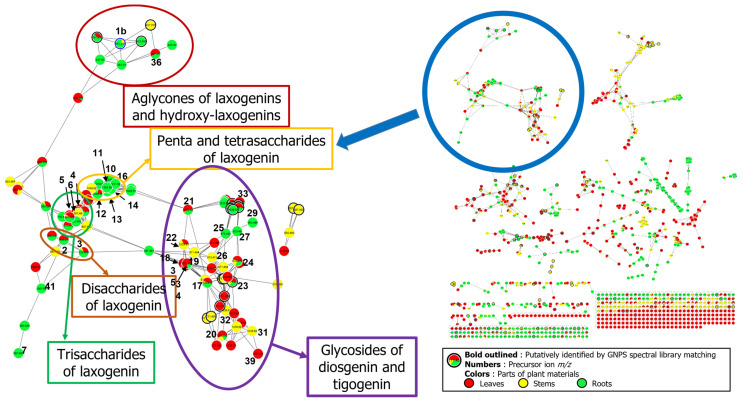
Comparative molecular network of laxogenin-(**1b**), diosgenin-(**1a**), and sarsasapogenin/tigogenin-(**1c**)-based saponins from leaf (red markers), stem (yellow markers), and root (green markers) materials of *S. sieboldii*.

**Figure 3 ijms-24-11487-f003:**
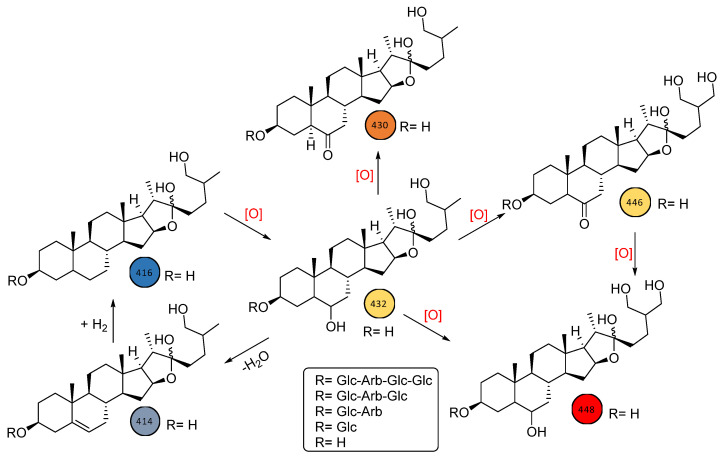
Biogenetic relationship between closely related oxidative furostanes within *S. sieboldii*.

**Table 1 ijms-24-11487-t001:** Accurate *m*/*z*, fragment ions of analytes in different parts (leaf, stem, and root) of *S. sieboldii* using UHPLC QToF.

#	Name	*t_R_* (min)	Molecular Formula	Mass	*m*/*z* [M-H]^−^	Error (ppm)	*m*/*z* [M+H]^+^	Error (ppm)	Fragment Ions (50 eV)	*S. sieboldii* Plant Part		
										Leaf	Stem	Root
***m*/*z* 431 as an aglycone**												
**1**	Laxogenin (Aglycone)	32.3	C_27_H_42_O_4_	430.3083	-	-	431.3161 (431.3156)	−1.2	413.3037 [M+H-H_2_O]^+^, 287.1998 [M+H-C_8_H_16_O_2_]^+^, 269.1894 [M+H-C_8_H_16_O_2_-H_2_O]^+^, 251.1789 [M+H-C_8_H_16_O_2_-2H_2_O]^+^	**+**	**+**	**+**
							453.2983 (453.2975) [M+Na]^+^	−1.8				
**2**	Unknown-1/2 (Smilaxin A/Laxogenin 3-*O*-*α*-L-arabinopyranosyl -(l→6)-*β*-D-glucopyranoside)	22.4	C_38_H_60_O_13_	724.4034	723.3958 (723.3961) **769.4012** **(769.4016)** **[M+HCOO]^−^**	0.4 0.6	725.4099 (725.4107)	1.1	593.3674 [M+H-Ara]^+^, 431.3159 [M+H-Ara-Glc]^+^, 413.3044 [M+H-Ara-Glc-H_2_O]^+^, 395.2931 [M+H-Ara-Glc-2H_2_O]^+^, 287.2000 [M+H -Ara-Glc-C_8_H_16_O_2_]^+^, 269.1889 [M+H-Ara-Glc-C_8_H_16_O_2_-H_2_O]^+^, 251.1784 [M+H-Ara-Glc-C_8_H_16_O_2_-2H_2_O]^+^	+	+	+
**3**		23.1								+	+	+
							**742.4376** **(742.4372) [M+NH_4_]^+^**	−0.5				
							747.3916 (747.3926) [M+Na]^+^	1.4				
**4**	Unknown-3–6 (Smilaxin B/Laxogenin 3-*O*-*β*-D-glucopyranosyl-(l→4)-*O*-[*α*-L-arabinopyranosyl-( l→6)]-*β*-D-glucopyranoside)	13.1	C_44_H_70_O_18_	886.4562	885.4483 (885.4489) 931.4554 (931.4544) [M+HCOO]^−^	0.7 −1.1	887.4646 (887.4635)	−1.3	725.4094 [M+H-Glc]^+^, 593.3676 [M+H-Glc-Ara]^+^, 431.3150 [M+H-2Glc-Ara]^+^, 413.3035 [M+H-2Glc-Ara-H_2_O]^+^, 287.1988 [M+H-2Glc-Ara-C_8_H_16_O_2_]^+^, 269.1883 [M+H-2Glc-Ara-C_8_H_16_O_2_-H_2_O]^+^, 251.1764 [M+H-2Glc-Ara-C_8_H_16_O_2_-2H_2_O]^+^	+	+	+
**5**		14.0								+	+	+
**6**		16.2								+	+	+
**7**		22.6					**904.4899** **(904.4900)** **[M+NH_4_]^+^**	0.2		+	+	+
**8**	Unknown-7/8 (26-*O*-*β*-D-glucopyranosyl-3*β*, 22ξ, 26-trihydroxy-(25R)-5α-furostan-6-one 3-*O*-*α*- L-arabinopyranosyl-(l→6)]-*β*-D-glucopyranoside)	12.9	C_44_H_72_O_19_	904.4668	903.4579 (903.4595) 949.4645 (949.465) [M+HCOO]^−^	1.8 0.5	**887.4628** **(887.4635)** **[M+H-H_2_O]^+^**	0.8	755.4205 [M+H-H_2_O-Ara]^+^, 725.4087 [M+H-H_2_O-Glc]^+^, 593.3677 [M+H-H_2_O-Glc-Ara]^+^, 431.3157 [M+H-H_2_O-2Glc-Ara]^+^, 413.3051 [M+H -2Glc-Ara-2H_2_O]^+^, 287.2002 [M+H-H_2_O-2Glc-Ara-C_8_H_16_O_2_]^+^, 269.1894 [M+H-2Glc-Ara-C_8_H_16_O_2_-2H_2_O]^+^, 251.1790 [M+H-2Glc-Ara-C_8_H_16_O_2_-3H_2_O]^+^	+	+	+
**9**		13.4								+	+	+
							927.4570 (927.456) [M+Na]^+^	−1.1				
**10**	Unknown-9–11 (26-*O-β*-D-glucopyranosyl- 3β, 22ξ, 26-trihydroxy-(25R)-5α-furostan-6-one 3-*O-β*-D-glucopyranosyl-(l→4)-*O*-[ *α*- L-arabinopyranosyl-(l→6)]-*β*-D-glucopyranoside)	12.2	C_50_H_82_O_24_	1066.5196	1065.5112 (1065.5123) 1111.5167 (1111.5178) [M+HCOO]**^−^** 578.2554 (578.258) [M+2HCOO]^2−^	1.0 1.0 4.9	**1049.5184** **(1049.5163)** **[M+H-H_2_O]^+^**	−2.0	917.4723 [M+H-H_2_O-Ara]^+^, 887.4619 [M+H-H_2_O-Glc]^+^, 725.4108 [M+H-H_2_O-2Glc]^+^, 593.3680 [M+H-H_2_O-2Glc-Ara]^+^, 431.3162 [M+H-H_2_O-3Glc-Ara]^+^, 413.3046 [M+H-3Glc-Ara-2H_2_O]^+^, 395.2945 [M+H-3Glc-Ara-3H_2_O]^+^, 287.2005 [M+H-H_2_O-3Glc-Ara-C_8_H_16_O_2_]^+^, 269.1898 [M+H-3Glc-Ara-C_8_H_16_O_2_-2H_2_O]^+^, 251.1784 [M+H-3Glc-Ara-C_8_H_16_O_2_-3H_2_O]^+^	ND	ND	+
**11**		12.8					1084.5538 (1084.5544) [M+NH_4_]^+^	−0.4		+	+	+
**12**		13.13					1089.5085 (1089.5088) [M+Na]^+^	0.3		+	+	+
**13**	Unknown-12	12.3	C_55_H_90_O_28_	1198.5619	1197.5523 (1197.5546)	1.9	**1181.5577** **(1181.5586)** **[M+H-H_2_O]^+^**	0.7	1067.5246 [M+H-Ara]^+^, 1049.5155 [M+H-H_2_O-Ara]^+^, 917.4684 [M+H-H_2_O-2Ara]^+^, 887.4642 [M+H-H_2_O-Ara -Glc]^+^, 755.4211 [M+H-H_2_O-2Ara -Glc]^+^, 593.3675 [M+H-H_2_O-2Ara -2Glc]^+^, 431.3149 [M+H-H_2_O-2Ara -3Glc]^+^, 413.3045 [M+H-2Ara -3Glc-2H_2_O]^+^, 395.2945 [M+H-2Ara -3Glc-3H_2_O]^+^, 287.2002 [M+H-H_2_O-3Glc-2Ara-C_8_H_16_O_2_]^+^, 269.1876 [M+H-3Glc-2Ara-C_8_H_16_O_2_-2H_2_O]^+^, 251.1793 [M+H-3Glc-2Ara-C_8_H_16_O_2_-3H_2_O]^+^	ND	+	+
							1216.5950 (1216.5957) [M+NH_4_]^+^	0.6				
							1221.5507 (1221.5511) [M+Na]^+^	0.3				
**14**	Unknown-13–15	11.6	C_56_H_92_O_29_	1228.5724	1227.5639 (1227.5652)	1.0	**1211.5693** **(1211.5691)** **[M+H-H_2_O]^+^**	−0.1	1049.5142 [M+H-H_2_O-Glc]^+^, 917.4690 [M+H-H_2_O-Glc-Ara]^+^, 887.4606 [M+H-H_2_O-2Glc]^+^, 755.4214 [M+H-H_2_O-2Glc-Ara]^+^, 593.3672 [M+H-H_2_O-3Glc-Ara]^+^, 431.3149 [M+H-H_2_O-4Glc-Ara]^+^, 413.3048 [M+H-3Glc-Ara-2H_2_O]^+^, 287.1995 [M+H-H_2_O-3Glc-Ara-C_8_H_16_O_2_]^+^, 269.1887 [M+H-3Glc-Ara-C_8_H_16_O_2_-2H_2_O]^+^, 251.1788 [M+H-3Glc-Ara-C_8_H_16_O_2_-3H_2_O]^+^	+	+	+
**15**		12.4								+	+	+
**16**		12.7								+	+	+
***m*/*z* 415 as an aglycone**												
**17**	Unknown-16	26.05	C_38_H_60_O_12_	708.4085	707.4010 (707.4012) 753.4058 (753.4067) [M+HCOO]^−^	0.3 1.2	709.4170 (709.4158)	−1.8	577.3738 [M+H-Ara]^+^, 415.3204 [M+H-Glc-Ara]^+^, 397.3092 [M+H-Glc-Ara-H_2_O]^+^, 379.3018 [M+H-Glc-Ara-2H_2_O]^+^, 271.2044 [M+H-H_2_O-Glc-Ara- C_8_H_16_O_2_]^+^, 253.1945 [M+H-Glc-Ara C_8_H_16_O_2_-2H_2_O]^+^	+	+	+
							726.4410 (726.4423) [M+NH_4_]^+^	1.8				
							731.3966 (731.3977) [M+Na]^+^	1.6				
**18**	Unknown-17/18	15.2	C_44_H_72_O_18_	888.4719	887.4642 (887.4646) 933.4690 (933.4701) [M+HCOO]^−^	0.4 1.2	**871.4686** **(871.4686)** **[M+H-H_2_O]^+^**	0	739.4221 [M+H-H_2_O-Ara]^+^, 709.4157 [M+H-H_2_O-Glc]^+^, 577.3738 [M+H-H_2_O-Glc-Ara]^+^, 415.3204 [M+H-H_2_O-2Glc-Ara]^+^, 397.3097 [M+H-2Glc-Ara-2H_2_O]^+^, 379.2989 [M+H-2Glc-Ara-3H_2_O]^+^, 271.2044 [M+H-H_2_O-2Glc-Ara-C_8_H_16_O_2_]^+^, 253.1950 [M+H-2Glc-Ara-C_8_H_16_O_2_-2H_2_O]^+^	+	+	+
**19**		15.4								+	+	+
**20**	Unknown-19–21	14.2	C_50_H_82_O_23_	1050.5247	1049.5158 (1049.5174) 1095.5219 (1095.5229) [M+HCOO]^−^	1.5 0.9	**1033.5236** **(1033.5214)** **[M+H-H_2_O]^+^**	−2.1	901.4810 [M+H-H_2_O-Ara]^+^, 871.4686 [M+H-H_2_O-Glc]^+^, 739.4254 [M+H-H_2_O-Glc-Ara]^+^, 709.4142 [M+H-H_2_O-2Glc]^+^, 577.3738 [M+H-H_2_O-2Glc-Ara]^+^, 415.3204 [M+H-H_2_O-3Glc-Ara]^+^, 397.3087 [M+H-3Glc-Ara-2H_2_O]^+^, 379.2985 [M+H-3Glc-Ara-3H_2_O]^+^, 271.2042 [M+H-H_2_O-3Glc-Ara-C_8_H_16_O_2_]^+^, 253.1945 [M+H-3Glc-Ara-C_8_H_16_O_2_-2H_2_O]^+^	t	+	t
**21**		14.6								+	+	+
**22**		14.8								+	+	+
							1073.5174 (1073.5139) [M+Na]^+^	−3.3				
***m*/*z* 417 as an aglycone**												
**23**	Unknown-22	26.5	C_38_H_62_O_12_	710.4241	709.4163 (709.4169) 755.4212 (755.4223) [M+HCOO]^−^	0.8 1.6	**711.4299** **(711.4314)**	2.1	579.3888 [M+H-Ara]^+^, 417.3317 [M+H-Glc-Ara]^+^, 399.3288 [M+H-Glc-Ara-H_2_O]^+^, 273.2201 [M+H-Glc-Ara-C_8_H_16_O_2_]^+^, 255.2095 [M+H-Glc-Ara-C_8_H_16_O_2_-H_2_O]^+^	+	+	+
							733.4133 (733.4133) [M+Na]^+^	0.0				
**24**	Unknown-23–25 (Smilaxin C/Tigogenin 3-*O*-*β*-D-glucopyranosyl-(l→4)-*O*-[*α*-L-arabinopyranosyl-(l→6)]-*β*-D-glucopyranoside)	15.6	C_44_H_72_O_17_	872.477	871.4688 (871.4697) 917.4753 (917.4752) [M+HCOO]^−^	1.0 −0.2	873.4834 (873.4842)	1.0	741.4422 [M+H-Ara]^+^, 711.4335 [M+H-Glc]^+^, 579.3789 [M+H-Glc-Ara]^+^, 417.3346 [M+H-2Glc-Ara]^+^, 399.3222 [M+H-2Glc-Ara-H_2_O]^+^, 273.2186 [M+H-2Glc-Ara-C_8_H_16_O_2_]^+^, 255.2092 [M+H-2Glc-Ara-C_8_H_16_O_2_-H_2_O]^+^	+	+	+
**25**		15.1								+	+	+
							**895.4652** **(895.4662)** **[M+Na]^+^**	1.1				
**26**		25.4								+	+	+
**27**	Unknown-26/27	15.5	C_44_H_74_O_18_	890.4875	889.4802 (889.4802) 935.4853 (935.4857) [M+HCOO]^−^	0.0 0.5	**873.4834** **(873.4842)** **[M+H-H_2_O]^+^**	1.0	711.4335 [M+H-H_2_O-Glc]^+^, 579.3789 [M+H-H_2_O-Glc-Ara]^+^, 417.3346 [M+H-H_2_O-2Glc-Ara]^+^, 399.3222 [M+H-2Glc-Ara-2H_2_O]^+^, 273.2186 [M+H-H_2_O-2Glc-Ara-C_8_H_16_O_2_]^+^, 255.2092 [M+H-2Glc-Ara-C_8_H_16_O_2_-2H_2_O]^+^	+	++	+
**28**		15.6								+	++	+
**29**	Unknown-28/29 (Sarsaparilloside *C*-furostane ring)	16.1	C_45_H_76_O_19_	920.4981	919.4906 (919.4908) 965.4931 (965.4963) [M+HCOO]^−^	0.2 3.5	**903.4943** **(903.4948)** **[M+H-H_2_O]^+^**	0.6	741.4372 [M+H-H_2_O-Glc]^+^, 579.3840 [M+H-H_2_O-2Glc]^+^, 417.3341 [M+H-H_2_O-3Glc]^+^, 399.3239 [M+H-3Glc-2H_2_O]^+^, 381.3138 [M+H-3Glc-3H_2_O]^+^, 273.2214 [M+H-H_2_O-3Glc -C_8_H_16_O_2_]^+^, 255.2099 [M+H-3Glc-C_8_H_16_O_2_-2H_2_O]^+^	+	+	+
**30**		16.2								+	+	+
**31**	Unknown-30–32 (Furostane-3,22,26-triol 3-*O*-[*α*-L-Arabinopyranosyl-(1→6)-[*β*-D-glucopyranosyl-(1→4)]-*β*-D-glucopyranoside], 26-*O*-*β*-D-glucopyranoside)	14.4	C_50_H_84_O_23_	1052.5403	1051.5333 (1051.5331) 1097.5389 (1097.5385) [M+HCOO]^−^	−0.2 −0.3	**1035.5391** **(1035.5371)** **[M+H-H_2_O]^+^**	−2.0	903.4945 [M+H-H_2_O-Ara]^+^, 873.4851 [M+H-H_2_O-Glc]^+^, 741.4401 [M+H-H_2_O-Glc-Ara]^+^, 729.3691 [M+H-Glc-Ara-2H_2_O]^+^, 711.4283 [M+H-H_2_O-2Glc]^+^, 579.3891 [M+H-H_2_O-2Glc-Ara]^+^, 417.3357 [M+H-H_2_O-3Glc-Ara]^+^, 399.3240 [M+H-3Glc-Ara-2H_2_O]^+^, 381.3159 [M+H-3Glc-Ara-3H_2_O]^+^, 273.2211 [M+H-H_2_O-3Glc-Ara-C_8_H_16_O_2_]^+^, 255.2098 [M+H-3Glc-Ara-C_8_H_16_O_2_-2H_2_O]^+^	+	+	ND
**32**		14.9								+	+	+
**33**		15.03								+	+	+
***m*/*z* 433 as an aglycone**												
**34**	Unknown-33/34	16.4	C_44_H_72_O_18_	888.4719	887.4653 (887.4646) 933.4685 (933.4701) [M+HCOO]^−^	−0.8 1.8	889.4792 (889.4791)	−0.1	757.4344 [M+H-Ara]^+^, 727.4261 [M+H-Glc]^+^, 595.3829 [M+H-Glc-Ara]^+^, 433.3306 [M+H-2Glc-Ara]^+^, 415.3190 [M+H-2Glc-Ara-H_2_O]^+^, 397.3067 [M+H-2Glc-Ara-2H_2_O]^+^, 255.2096 [M+H-2Glc-Ara-2H_2_O-C_8_H_14_O_2_]^+^	t	t	+
**35**		20								+	t	+
							911.4622 (911.4611) [M+Na]^+^	−1.3				
***m*/*z* 447 as an aglycone**												
**36**	Unknown-35 (Sieboldigenin, Aglycone)	23.7	C_27_H_42_O_5_	446.3032	-	-	**447.3112** **(447.3105)**	−1.6	429.2996 [M+H-H_2_O]^+^, 287.1993 [M+H-H_2_O-C_8_H_14_O_2_]^+^, 269.1882 [M+H-C_8_H_14_O_2_-2H_2_O]^+^, 251.1775 [M+H-C_8_H_14_O_2_-3H_2_O]^+^	ND	t	+
							469.2921 (469.2924) [M+Na]^+^	0.8				
**37**	Unknown-36/37 (Sieboldin B)	16.7	C_38_H_60_O_14_	740.3938	739.3901 (739.391) 785.3966 (785.3965) [M+HCOO]^−^	1.3 −0.1	741.4063 (741.4056)	−1.0	609.3629 [M+H-Ara]^+^, 579.3542 [M+H-Glc]^+^, 447.3106 [M+H-Ara-Glc]^+^,	+	+	+
**38**		17.4					763.3873 (763.3875) [M+Na]^+^	0.3	609.3626 [M+H-Ara]^+^, 447.3097 [M+H-Ara-Glc]^+^, 429.2990 [M+H-Ara-Glc-H_2_O]^+^, 411.2890 [M+H-Ara-Glc-2H_2_O]^+^, 287.1995 [M+H-Ara-Glc-H_2_O-C_8_H_14_O_2_]^+^, 269.1892 [M+H-Ara-Glc-2H_2_O-C_8_H_14_O_2_]^+^, 251.1785 [M+H-Ara-Glc-3H_2_O-C_8_H_14_O_2_]^+^	+	+	+
**39**	Unknown-38/39 (Sieboldin A-(3*β*, 27-dihydroxy-(25S)-5α-spirostan-6-one 3-*O*-*β*-D-glucopyranosyl-(l→4)-*O*-[*α*-L-arabinopyranosyl-(l→6)]-*β*-D-glucopyranoside))	16.13	C_44_H_70_O_19_	902.4511	901.4435 (901.4439) 947.4495 (947.4493) [M+HCOO]^−^	0.4 −0.2	903.4536 (903.4584)	5.3	771.4178 [M+H-Ara]^+^, 741.4070 [M+H-Glc]^+^, 609.3633 [M+H-Glc-Ara]^+^, 579.3526 [M+H-2Glc]^+^, 447.3099 [M+H-2Glc-Ara]^+^, 429.3001 [M+H-2Glc-Ara-H_2_O]^+^, 287.1999 [M+H-2Glc-Ara-H_2_O-C_8_H_14_O_2_]^+^	+	+	+
							925.4407 (925.4404) [M+Na]^+^	−0.4				
**40**		16.8								+	+	+
***m*/*z* 449 as an aglycone**												
**41**	Unknown-40 (tetra-glycoside of 3, 6, 27-trihydroxy furostane-type saponin)	12.3	C_50_H_84_O_25_	1084.5302	1083.5229 (1083.5229) 1129.5257 (1129.5284) [M+HCOO]^−^	0.0 2.5	**1067.5268** **(1067.5269)** **[M+H-H_2_O]^+^**	0.1	935.4892 [M+H-H_2_O-Ara]^+^, 905.4713 [M+H-H_2_O-Glc]^+^, 773.4279 [M+H-H_2_O-Glc-Ara]^+^, 743.4184 [M+H-H_2_O-2Glc]^+^, 725.4059 [M+H-2Glc-2H_2_O]^+^, 611.3740 [M+H-H_2_O-2Glc-Ara]^+^, 593.3687 [M+H-2Glc-Ara-2H_2_O]^+^, 449.3214 [M+H-H_2_O-3Glc-Ara]^+^, 431.3148 [M+H-3Glc-Ara-2H_2_O]^+^, 413.3029 [M+H-3Glc-Ara-3H_2_O]^+^, 395.2903 [M+H-3Glc-Ara-4H_2_O]^+^, 287.1994 [M+H-3Glc-Ara-2H_2_O-C_8_H_16_O_2_]^+^, 269.1873 [M+H-3Glc-Ara-3H_2_O-C_8_H_16_O_2_]^+^, 251.1770 [M+H-3Glc-Ara-4H_2_O-C_8_H_16_O_2_]^+^	+	+	+

Underlined numbers are indicated as base peaks; bolded ones are shown in major; t = trace; ND = not detected; + = detected; ++ = detected with good amount.

## Data Availability

Not applicable.

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
