# Peer review of "6-Oxofurostane and (iso)Spirostane Types of Saponins in *Smilax sieboldii*: UHPLC-QToF-MS/MS and GNPS-Molecular Networking Approach for the Rapid Dereplication and Biodistribution of Specialized Metabolites"

_ijms, 2023, doi:10.3390/ijms241411487_

Round 1

Reviewer 1 Report (Previous Reviewer 2)

Bharathi Avula et al. performed an UHPLC-QToF-MS/MS-based chemical study on the leaf, stem, and root/rhizome of Smilax sieboldii, expanded with GNPS platform, to explore the content of saponins, especially 6-oxofurostane and (iso)spirostane. The manuscript has relevant and attractive elements for readers, and the authors addressed the comments performed on the previous version. So, the manuscript improved in quality and content. Only some minor points remain to be addressed.

1.       Title: “6-Oxofurostane” instead of “6-Oxo furostane” (without spacing)

2.       Lines 30-33: Improve the meaning of this relevant sentence. For instance, it is not clear what is the meaning of “were experimental” within the context of this sentence.

3.       Line 179: Smilax spp. in italics.

4.       Section 2.2.: Important to clarify how the authors could establish unequivocally the epimeric forms of the sugars (i.e., α o β) of the annotated glycosides (e.g., compounds 2,3,6,7,9,11, among others) since the measurements were performed by HRMS/MS and the epimeric forms can not be unequivocally established. Therefore, the adequate scope and even the strategy must be strictly informed and written in this section to avoid confusing interpretations by readers. Identical considerations must be informed for the identified aglycones. Be consistent throughout the manuscript.

Detailed scrutiny should be performed throughout the manuscript to revise grammar and stylistic issues.

Author Response

Response to Reviewer 1 Comments

Dear Reviewer,

Thank you for your invaluable feedback and suggestions on our revised manuscript on “6-Oxofurostane and (iso)spirostane type of saponins in Smilax sieboldii: UHPLC-QToF-MS/MS and GNPS-molecular networking approach for the rapid dereplication and biodistribution of specialized metabolites“. Following this response form, we will provide information on the changes we have implemented in response to your feedback:

  1. Title: “6-Oxofurostane” instead of “6-Oxo furostane” (without spacing)

Response: The title has been changed as suggested.

  1. Lines 30-33: Improve the meaning of this relevant sentence. For instance, it is not clear what is the meaning of “were experimental” within the context of this sentence.

Response: The sentence has been corrected and now reads as “Tandem mass diagnostic fragmentation patterns of aglycones, diosgenin, sarsasapogenin/tigogenin, or laxogenin were critical to establishing the unique nodes belonging to six groups of nineteen unknown steroidal saponins identified in S. sieboldii.”.

  1. Line 179: Smilax in italics.

Response: Corrected.

  1. Section 2.2.: Important to clarify how the authors could establish unequivocally the epimeric forms of the sugars (i.e., α o β) of the annotated glycosides (e.g., compounds 2,3,6,7,9,11, among others) since the measurements were performed by HRMS/MS and the epimeric forms can not be unequivocally established. Therefore, the adequate scope and even the strategy must be strictly informed and written in this section to avoid confusing interpretations by readers. Identical considerations must be informed for the identified aglycones. Be consistent throughout the manuscript.

Response: Thank you for the critical comment. As the reviewer pointed out, the epimeric forms cannot be established by HRMS unequivocally. We completely agree with the reviewer and added the following sentence before cluster analysis 2.2.1 to avoid possible misinterpretation solely based on mass spectral data. We intentionally added the word ‘tentatively’ throughout the manuscript as part of further revisions.

“Based on these mass spectral features, at least five unique clusters were identified (as outlined below), in which several isobaric, anomeric isomers were identified tentatively and compared with reported data if applicable [8].”

Comments on the Quality of English Language:

Detailed scrutiny should be performed throughout the manuscript to revise grammar and stylistic issues.

Response: As suggested, several grammatical changes, including stylistic, were incorporated as part of this round of revisions.

We sincerely hope that our efforts in responding to your comments have been effective. If you have any further suggestions for improvement or concerns, please do not hesitate to let us know. We greatly appreciate your expertise and guidance.

Sincerely yours, 

Ji-Yeong Bae, Ph.D.

Assistant Professor

College of Pharmacy

Jeju National University

Reviewer 2 Report (Previous Reviewer 1)

Manuscript by Avula et al. concerns the analysis of different parts of plant Smilax sieboldii; the content was established by mass-spectroscopy (MS). I reviewed the previous version of this article, and found that the content was changed significantly (but not fully) in accordance with my Notes. Thus, the section “Materials and Methods” is absent; so, it is impossible to reproduce the results obtained. I think that it is necessary to add a short description of instruments, the way of samples preparation and the analysis of data (including methods for graphics construction etc.).

Minor editing of English language required.

Author Response

Response to Reviewer 2 Comments

Dear Reviewer,

We would like to express our sincere appreciation for your valuable feedback and constructive comments on our manuscript entitled “6-Oxofurostane and (iso)spirostane type of saponins in Smilax sieboldii: UHPLC-QToF-MS/MS and GNPS-molecular networking approach for the rapid dereplication and biodistribution of specialized metabolites“. We are grateful for the time and effort you have taken to examine our work and to give suggestions for improving the quality and clarity of our research. 

Manuscript by Avula et al. concerns the analysis of different parts of plant Smilax sieboldii; the content was established by mass-spectroscopy (MS). I reviewed the previous version of this article, and found that the content was changed significantly (but not fully) in accordance with my Notes. Thus, the section “Materials and Methods” is absent; so, it is impossible to reproduce the results obtained. I think that it is necessary to add a short description of instruments, the way of samples preparation and the analysis of data (including methods for graphics construction etc.).

Response: As suggested, the requested information on materials and methods, including sample preparation, instrument details, and network analysis, are included in Section #3.

Comments on the Quality of English Language:

Minor editing of English language required.

Response: As suggested, several grammatical changes, including stylistic, were incorporated as part of this round of revisions.

We are certain that these revisions have greatly strengthened our manuscript and brought clarity to the readers Thank you once again for your thoughtfulness.

Sincerely yours,

 Ji-Yeong Bae, Ph.D.

Assistant Professor

College of Pharmacy

Jeju National University

This manuscript is a resubmission of an earlier submission. The following is a list of the peer review reports and author responses from that submission.

Round 1

Reviewer 1 Report

Manuscript by Avula et al. concerns the analysis of different parts of plant Smilax sieboldii; the content was established by mass-spectroscopy (MS). It was established, that various saponins are the main compounds. Authors used MS without any preprocessing, making their approach simple and useful as an express method. Several important aspects of modern science are covered, including not only MS, but phytochemistry, biosynthesis also. The Manuscript may be regarded as multidisciplinary, interesting to wide readership, to researchers working in biology, organic chemistry, mass-spectrometry, natural compounds. I found this Manuscript interesting and suitable for publication in Int. J. Mol. Sci. At the same time several questions are appeared after reading. Thus, there is no “Methods and Materials” section, content of Table 1 is unreadable etc. (see below). I propose Major Revision before possible acceptance.

Several Notes:

11)      Add List of abbreviation.

22)      Phrases “…the MS/MS” (Abstract, page 12 etc.) are confusing.

33)      Page 3, sentence “… the proposed molecular formulas and possible compounds” looks strange. Are you completely sure of the reliability of your interpretation?

44)      Table 1: the content is unreadable. Change the width of columns.

55)      Check references 6, 8.

May be improved (Minor polishing).

Reviewer 2 Report

The manuscript ID ijms-2401313 describes the phytochemical study of the leaf, stem, and root/rhizome of Smilax sieboldii using UHPLC-QToF-MS/MS and GNPS platforms. The manuscript is very interesting and involves important information for readership. However, some issues should be addressed before being considered further.

1.      The abstract can be shortened to be more comprehensive since it is challenging to be followed.

2.      Table 1 is too large (four pages), so it can be summarized and condensed, and the full one might be included as supplementary material.

3.      The quality and size of Figures 2 to 7 must be improved since the information is challenging to be visualized.

4.      Line 377: was the sonication effect on the metabolite integrity verified to avoid artifact formation? This should be clarified in the manuscript.

5.      Detailed scrutiny of the accurate mass values in section 2.2 should be performed, looking for misleading and discrepancies between text and figures.

6.      The discussion is highly descriptive and seems to be an extension of the results section, so the discussion can be improved since no references or comparisons to previous reports are provided.

7.      The HRMS spectra (positive and negative mode) of the 41 identified metabolites must be provided in the supplementary material.